# Sample-Efficient and Statistically Rigorous Policy Comparison Using General Robot Performance Measures

Authors omitted for anonymized review.*

*Abstract*— **Robot manipulation policies are generally limited in evaluation to a small number of hardware rollouts. This strong resource constraint necessitates reliable, sample-efficient evaluation procedures to properly assess model capabilities, including robustness to environment variation. This work presents N-SCORE, an efficient, rigorous procedure to certify policy comparisons for general performance measures. Based on safe, anytime-valid inference (SAVI), our *sequential* approach allows the evaluator to *stop early* when sufficient statistical evidence has accumulated to reach a decision at a pre-specified level of confidence. Importantly, N-SCORE applies to *nonparametric settings* representing *complex behavioral indicators* significantly generalizing prior work. We empirically demonstrate significant reduction in evaluation burden versus sequential procedures designed for less informative measures, with no loss of statistical rigor. These results are suggestive of significant potential savings for future evaluation of general robustness measures.**

## I. Introduction

Advances in robot policy synthesis incorporate increasing complexity throughout the design process, training on large-scale datasets, using sophisticated, stochastic architectures, and requiring commensurate increases in training resources. These advances have led to significant improvements in solving dexterous and long-horizon tasks [1], inferring semantic information from context [2], and safely interacting with numerous other autonomous agents [3]. However, this complexity makes design decisions like the choice of dataset, the training or fine-tuning procedure, and the network architecture analytically opaque. The value of novel interventions cannot be proven from first principles, requiring rigorous analysis [4], [5] of effects on empirical performance.

Unfortunately, such analysis poses a significant challenge in the robotics setting. Hardware evaluation is expensive and slow to collect, while evaluation in simulation continues to suffer from sim-to-real gaps [6]. When this can be overcome, procedures are often still limited to coarse or highly limited classes of evaluation metrics [7], [8]. For example, binary measurement of task success or failure, the *de facto* standard for measuring the performance of robot manipulation policies (particularly on hardware), can obscure valuable information about the robot behavior [5], [9]. For metrics of manipulation robustness that extend beyond high empirical success or task completion, a sample-efficient, statistically rigorous evaluation procedure that adapts to *general behavioral metrics* is necessary to reliably certify improvement, and thus to measure and track progress towards this goal.

*An extended version of this work is currently under review in a separate, archival robotics venue.

This work presents **N**onparametric **S**equential **CO**mparison for **R**igorous **E**valuation (N-SCORE) to compare general measures of policy performance. N-SCORE (see Figure 1) addresses the shortcomings of prior approaches by explicitly and simultaneously accounting for three desiderata: statistical rigor, sample efficiency, and applicability to general performance and robustness measures. We summarize our **contributions** as follows:

1) We introduce N-SCORE, a novel sequential policy comparison method for general performance measures;
2) We prove that N-SCORE is statistically rigorous in the sense of controlling Type-1 Error, and empirically demonstrate its improved sample efficiency compared to state-of-the-art (SOTA) baselines;
3) We comprehensively evaluate N-SCORE on large, high-impact evaluation datasets in robotics comprising over 4500 hardware evaluation rollouts and 2000 high-fidelity simulation rollouts [1], [10], demonstrating significant savings in requisite evaluation effort to rigorously certify policy improvements.

## II. Preliminaries

Due to space constraints, we defer related work to App. I and proceed to the technical exposition. We model a robot that must complete a task while subject to stochastic uncertainty in its environment as a Partially Observable Markov Decision Process (POMDP) [11]. We assume the existence (potentially, specification) of a real-valued scalar performance (robustness) measure $R$ encoding the success or robustness the robot demonstrates in completing the task instance (or generalizing across some measure of uncertainty or variation). Conditioned on an arbitrary robot policy $\pi$, the environment uncertainty is compressed to uncertainty over outcomes through a policy rollout to a (possibly complicated) distribution over the performance measure $\mathcal{D}_R$. Our goal is to make rigorous comparisons for as general a class of $\mathcal{D}_R$ as possible. With this in mind, we make a single restriction: that $R$ be bounded w.p. 1. Unless stated otherwise, we assume henceforth that $R$ is a bounded performance measure but make no other structural assumptions. We are then tasked with quickly and accurately comparing the means of two distributions (for the baseline and the novel policy) over the measure $R$:

$$\text{Test: } \mathbb{E}_{r \sim \mathcal{D}_R^{[0]}}[r] < \mathbb{E}_{r \sim \mathcal{D}_R^{[1]}}[r].$$

This encompasses many practical problems, for example: comparing two policies with different architectures or trained on different data ($\pi_0$ and $\pi_1$), comparing distribution shift in

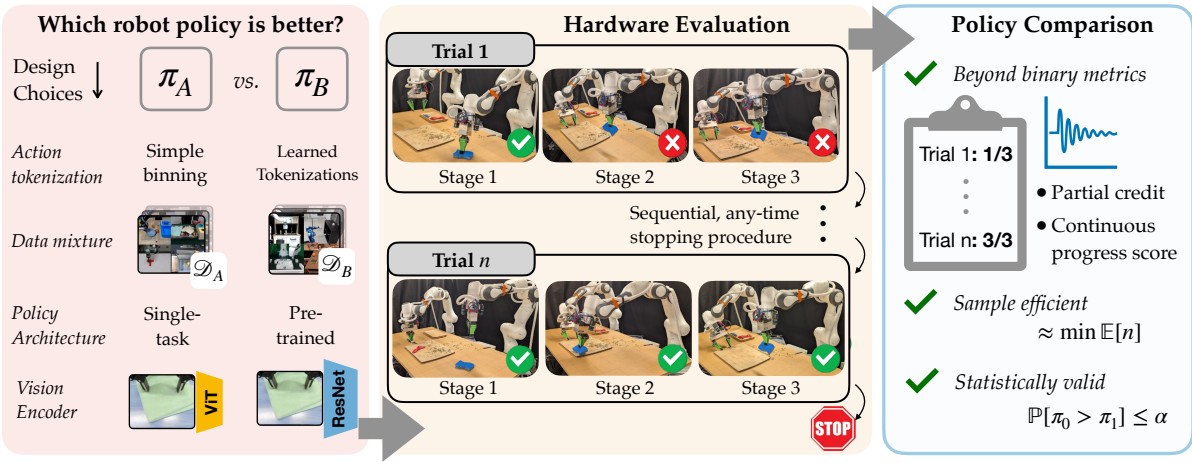

Fig. 1: The evaluation context of the N-SCORE procedure. (Left) We consider the problem of policy comparison for general performance measures, which arises out of counterfactual design decisions in the policy synthesis process (e.g., for the purpose of increasing the measure of policy robustness). (Middle) Evaluation on hardware or in high-fidelity simulation is the gold standard to assess the effect of such changes, but is costly to collect. (Right) N-SCORE is a sequential evaluation procedure that is statistically rigorous, sample efficient, and generalizes to rich, diverse measures of robot performance and robustness.

task realizations for a single policy, or comparing the effects of different hyperparameters in training. Importantly, because the true means are unknown and not directly observable, a testing decision must be made on the basis of empirical performance in evaluation. Due to stochastic uncertainty in gathering this data, decisions are inherently *probabilistic*. Thus, we seek statistical assurances bounding the probability of incorrect inferences:

$$
\begin{aligned}
&\text{If:} \quad \mathbb{E}_{\mathcal{D}_R^{[0]}}[R] \geq \mathbb{E}_{\mathcal{D}_R^{[1]}}[R] \\
&\text{Then:} \ \mathbb{P}\left[\text{Wrongly decide } \mathbb{E}_{\mathcal{D}_R^{[0]}}[R] < \mathbb{E}_{\mathcal{D}_R^{[1]}}[R]\right] \leq \alpha^*.
\end{aligned} \tag{1}
$$

The challenges to obtaining a result in the form of Equation (1) are twofold. First, the decision must be statistically rigorous—it must safeguard against inadvertently reporting statistical noise as genuine improvement. This can require a substantial number of evaluations when the difference is small or the desired confidence $1-\alpha^*$ is very high. However, the result also must be obtained as quickly as possible, so that the evaluator can efficiently allocate hardware or computational resources to begin investigating other questions. As noted elsewhere [7], allocating limited evaluation resources is a major bottleneck in improving policy performance. The sequential nature of N-SCORE balances statistical confidence and speed of decision-making, adaptively making decisions when precisely enough evidence has accumulated and minimizing unnecessary evaluation trials.

## III. PROBLEM FORMULATION

Section II introduced the policy comparison problem, which seeks guarantees as in Equation (1). The challenge in designing a test is the complexity of $\mathcal{D}_R^{[i]}$, as our procedure must be robust to *any performance (or robustness) measure*.

### A. The Formal Hypotheses

We formulate the problem in the context of frequentist statistical testing [12], constructing a general null (skeptical) hypothesis and an alternative (desired) hypothesis which reflect *all possible* cases of Equation (1). In this general *nonparametric setting* (see App. I-C), we define $\mathcal{M}_{[0,1]}$ to encompass all Lebesgue-measurable distributions on the real interval $[0, 1]$. We partition $\mathcal{M}_{[0,1]}$ into two parts:

$$
\begin{aligned}
S^- &:= \{(\mathcal{D}_R^{[0]}, \mathcal{D}_R^{[1]}) \in \mathcal{M}_{[0,1]}^2 : \mathbb{E}_{\mathcal{D}_R^{[0]}}[R] \geq \mathbb{E}_{\mathcal{D}_R^{[1]}}[R]\} \ \text{and} \\
S^+ &:= \{(\mathcal{D}_R^{[0]}, \mathcal{D}_R^{[1]}) \in \mathcal{M}_{[0,1]}^2 : \mathbb{E}_{\mathcal{D}_R^{[0]}}[R] < \mathbb{E}_{\mathcal{D}_R^{[1]}}[R]\}.
\end{aligned}
$$

Thus, our test formulates null and alternative hypotheses as $\mathcal{H}_0$ (resp., $\mathcal{H}_1$): $(\mathcal{D}_R^{[0]}, \mathcal{D}_R^{[1]}) \in S^-$ (resp. $S^+$).

Semantically, $\mathcal{H}_1$ (resp., $\mathcal{H}_0$) is understood to signify "all possible true states of the world in which the novel innovation is (resp., not) more robust than the baseline method." Equation (1) amounts to being $1 - \alpha^*$ confident that the true state of the world is *not* in $S^-$.

### B. Measuring the Quality of an Evaluation Algorithm

Given the hypotheses $\mathcal{H}_0$ and $\mathcal{H}_1$, we measure the quality of an evaluation procedure by two objectives: making decisions *quickly* and *correctly*. There are two types of errors that can be made. First, in the case that $\mathcal{H}_0$ is true, the evaluation algorithm may incorrectly decide that $\mathcal{H}_1$ is true. This is termed a false positive, or a 'Type-1 Error,' and its rate of occurrence is denoted $\alpha \in (0, 1)$ (see Equation (1)). Conversely, in the case that $\mathcal{H}_1$ is true, the protocol may incorrectly decide that $\mathcal{H}_0$ is true. This is a false negative, or a 'Type-2 Error,' and its rate is denoted $\beta \in (0, 1)$. These two error types combine to give a complete accounting of the algorithm's correctness. The last metric is the sample efficiency, denoted $\mathbb{E}[N]$. As presented, an optimal evaluation

algorithm — one that is quick and correct — minimizes all three of the measures $\{\alpha, \beta, \mathbb{E}[N]\}$.

### C. Efficient Tests as a Multi-Objective Optimization

There are fundamental tradeoffs which prevent simultaneous minimization of all three metrics. Therefore, we adopt the Neyman-Pearson testing approach [13], which normatively chooses to rigorously control the Type-1 Error rate, requiring that $\alpha \leq \alpha^*$ be treated as a hard constraint. The problem of designing an evaluation procedure then reduces to finding (near-)optimal decision rules to the following multi-objective optimization problem:

$$\gamma^* = \arg \inf_{\gamma \in \Gamma} \left[ \mathbb{E}[N](\gamma) + \lambda \cdot \beta(\gamma) \right] \quad (2)$$
$$\text{s.t. } \alpha(\gamma) \leq \alpha^*.$$

Here, $\Gamma$ denotes a space of decision-making rules (evaluation procedures) and $\lambda \in [0, \infty)$ is a nonnegative parameter trading off the time to decision and false negative rate. Efficiently solving Equation (2) results in an evaluation procedure that is guaranteed to maintain statistical rigor and quickly and effectively detect changes in performance. This provides the roboticist with the capacity to quickly iterate on new innovations, while ensuring justified and well-calibrated confidence in reported performance improvements.

## IV. METHODOLOGY

Any evaluation method must construct a process to distinguish $S^-$ from $S^+$. Intuitively, we will construct a scalar-valued dynamical system that behaves fundamentally differently when $\mathcal{H}_0$ is true than when $\mathcal{H}_1$ is true, based on the empirical evidence observed during evaluation. The testing problem then reduces to assessing certain stability properties of the dynamical system, made rigorous within the SAVI framework described in App. III.

### A. Constructing an 'Incremental Evidence Integrator'

A natural correlate to $\mathcal{H}_1$ is the empirical difference in the evaluation reward; for the $n^{th}$ evaluation trial, we consider:

$$\Delta \text{ evidence}_n \approx \left[ \xi \cdot (r_{1,n} - r_{0,n}) \right],$$

where $\xi > 0$ is a positive scaling factor and $r_{0,n}$, $r_{1,n}$ are the observed progress values. Intuitively, the evidence for $\mathcal{H}_1$ is positive when $r_{1,n} > r_{0,n}$ and negative otherwise. By assumed independence of the evaluation trials, this marginal evidence does not depend on $n$ (i.e., is time-invariant). Results in the SAVI literature (among other statistical estimation contexts) suggest that the optimal aggregation rate [14] is *multiplicative*, resulting in a measure of aggregate evidence $X_n$ (setting $X_0 = 1$ w.l.o.g.) that evolves according to:

$$X_{n+1} = (1 + \xi \cdot (r_{1,n} - r_{0,n}))X_n. \quad (3)$$

Therefore, when the evidence is positive for $\mathcal{H}_1$, the growth rate is greater than one, and the system is locally unstable. Conversely, when it is negative (i.e., in favor of $\mathcal{H}_0$), the growth rate is less than one and the system is stable. We

---

**Algorithm 1** N-SCORE Evaluation Protocol

**Input:**
Type-1 error limit $\alpha^* \in (0, 1)$, evaluation limit $N_{\max} > 0$.
**Initialize:**
$X_0 = 1$; $\bar{X} = 1$; $\xi_0 = 0$; $\mathcal{F}_0 = \{\emptyset\}$; $n = 1$.
**while** $\bar{X} < 1/\alpha^*$ **and** $n \leq N_{max}$ **do**
    Observe evaluation progress scores: $r_{0,n}$, $r_{1,n}$
    Compute increment: $a_{n-1} = 1 + \xi_{n-1}(r_{1,n} - r_{0,n})$
    Update martingale: $X_n \leftarrow a_{n-1} \cdot X_{n-1}$
    Update test statistic: $\bar{X} \leftarrow \max\{\bar{X}, X_n\}$
    Update filtration: $\mathcal{F}_n \leftarrow \mathcal{F}_{n-1} \cup (r_{0,n}, r_{1,n})$
    Update $\xi_n \leftarrow \text{proj}_{[0,1]} \, g(\mathcal{F}_n)$
    $n \leftarrow n + 1$
**end while**
**if** $\bar{X} < 1/\alpha^*$ **then**
    **return** Fail to Reject Null
**else**
    **return** Reject Null
**end if**

---

can additionally optimize $\xi_n = g(\mathcal{F}_{n-1})$ online based on the evidence accumulated so far as long as the choice of $\xi_n$ is independent of $(r_{0,n}, r_{1,n})$.

We designate an instability threshold as $1/\alpha^*$, for a desired Type-1 Error rate $\alpha^*$ of the test procedure:

$$\text{If:} \quad X_n \geq \frac{1}{\alpha^*};$$
$$\text{Then:} \left[ \text{Stop at step } n, \text{ and conclude } \mathcal{H}_1 \text{ is true.} \right]. \quad (4)$$

N-SCORE is operationalized in Algorithm 1. We show in Section IV-B that N-SCORE balances time-to-decision and correctness while rigorously controlling the Type-1 Error rate at tunable, pre-specified level $\alpha^*$. This enables rigorous decision-making yielding statements of the form of Equation (1) while minimizing the necessary evaluation burden.

### B. Theoretical Properties of Algorithm 1

We briefly outline the theoretical result for N-SCORE. Detailed proofs are deferred to App. III-B.

*Theorem 1 (Type-1 Error Control of Algorithm 1):*
Consider the N-SCORE evaluation procedure (Algorithm 1), utilizing the process defined in Equation (3). Then

$$\mathbb{P}\left[ \mathcal{H}_0 \text{ is true} \middle| \text{decide } \textbf{Reject Null} \right] \leq \alpha^* \quad (5)$$

Enforcing Type-1 Error control ensures high confidence (at level $1 - \alpha^*$) that reported significant innovations and improvements to the state-of-the-art are strictly separated from the inherent noise in robotic evaluation.

*Remark 1 (Efficient Optimization of $\xi_n$):* Consider the stochastic process family described in Equation (3), and let $\mathcal{F}_n$ be the natural filtration $\{(r_{0,i}, r_{1,i})\}_{i=1}^{n-1}$ at step $n$. There exists an efficient algorithm to maximize (online) over $\{\xi_n\}_{n=1}^N$ the expected growth rate of $\{X_n\}_{n=1}^N$. Intuitively, $\xi_n$ modulates the degree of confidence in marginal changes to $X_n$. Large $\xi_n$ induce faster growth when

data is favorable, but are penalized more harshly when it is not. Effectively modulating $\xi_n$ online is central to improving sample efficiency across many evaluation problems.

## V. EXPERIMENTAL RESULTS

We evaluate N-SCORE across a broad suite of evaluation settings, focusing herein upon the following research question: (**Informative Metrics**) *How effectively can N-SCORE detect significant differences for general, nonparametric evaluation scores (e.g., robustness measures)?* To answer this, we present results on synthetic nonparametric data and complex RL reward signals. Further empirical results, including on hardware data, are deferred to App. II.

### A. Baseline Methods

For brevity, we must defer detailed discussion of the baseline methods to App. I-D. Two key points of emphasis to situate the empirical results are: first, that existing SOTA evaluation procedures [7], [8] cannot be applied to general, nonparametric measures of robustness; second, general nonparametric approaches, like the WSR procedure [15], are not efficiently tailored to comparison problems, resulting in empirical inefficiency in these decision-making contexts.

|       | STEP | $\theta$-SAVI | N-SCORE | WSR   |
|-------|------|---------------|---------|-------|
| TTD   | –    | –             | 206.8   | 247.3 |
| Power | –    | –             | 0.889   | 0.840 |

TABLE I: **Mean time-to-decision (TTD) and power for each method on simulated nonparametric data.**

### B. Results on Simulated Nonparametric Data

We begin with results on simulated data generated from nonparametric densities. Random polynomials (order less than or equal to 10) are sampled and then rectified into an appropriate density via a shift-and-scale transformation to ensure nonnegativity on $[0, 1]$ and integration to 1. This process is analytically opaque, making it difficult to express the resulting family of distributions in any parametric form. Thus, only N-SCORE and WSR are applicable. We run 3000 comparison sequences of pairs of these distributions, limiting the gap in mean performance to be at least 0.01. For each sequence, up to $N = 1000$ evaluation scores are drawn from each distribution. Table I shows summary statistics for N-SCORE and WSR, respectively. Further, using N-SCORE as a meta-evaluator, we report that the difference in expected time-to-decision (206.8 vs. 247.3) is statistically significant at $\alpha = 0.05$, demonstrating that N-SCORE is significantly more sample efficient in this nonparametric context.

### C. Real-valued Continuous Metrics

The second example considered in the main text is ranking reinforcement learning (RL) policies based on continuous-valued, nonparametric episodic rewards. These should be understood as proxy cases for future robustness measures, which will likely reflect aggregated evidence from trajectory (state and action) data. Figure 2 shows the time-to-decision[1]

---

[1]Other tasks are included in App. II.

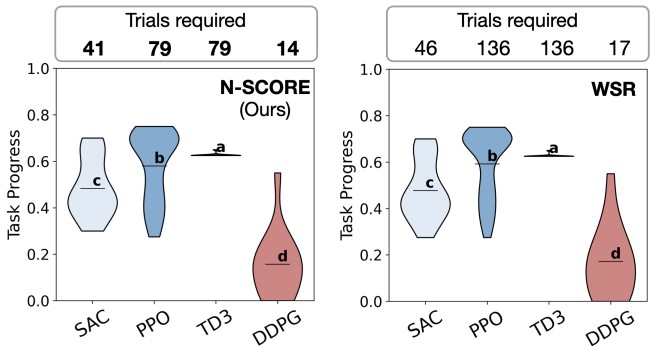

Fig. 2: **RL policy ranking (CLD plot) for Mujoco benchmark [Hopper].**

required to distinguish popular RL algorithms (PPO, TD3, DDPG, and SAC) on the Mujoco [16] benchmark Hopper task. Due to the continuous nature of the reward metric, only N-SCORE and WSR are applicable. The compact letter display (CLD) plot represents the reward distribution of the respective policies on new environment instances; the number of trials required to distinguish a policy from *all* others is included above the plot. The sample complexity of attaining a *full ranking* can be found by summation (for N-SCORE, 213; for WSR, 335).

## VI. DISCUSSION

We briefly situate the empirical results with respect to potential developmental avenues of robustness measures, mapping the applicability of N-SCORE to both present and future developments in this area.

A key challenge towards improving manipulation policy robustness is precisely one of measurement: how is robustness to be assessed? Many standard approaches are consistent with the use of N-SCORE. For example, increasing variation in object and environment datasets [10], [1] and assessing effects on downstream performance fits directly into the comparison framework. Separately, interventions to predict or detect failures [17], [18], [19] motivate the development of novel within-trajectory robustness signals like surprise metrics [20] or behavioral indicators like jitter [9]. Unlike, for example, partial credit success measures, complex within-trajectory indicators of policy robustness will likely be highly complex and nonparametric, *necessitating* the use of carefully chosen tools to distinguish candidate state-of-the-art innovations from statistical noise.

## VII. CONCLUSION

We have introduced and validated N-SCORE, a novel procedure for statistically precise sequential comparison of robot policies. N-SCORE generalizes to complex performance measures with state-of-the-art sample efficiency. The promise of such methods is to empower careful and rigorous codification of robot performance across a broad spectrum of performance indicators. As such, N-SCORE complements the development of indicators and metrics of manipulation policy robustness while providing a framework to measure and benchmark progress within the field.

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

Efficient evaluation is increasingly appreciated as a critical aspect of the robot policy design process. The fundamental concern of these approaches is the strict limit on available hardware data. A core focus lies in constructing proxy signals (e.g., using simulations or evaluator preferences) to circumvent this constraint. In parallel, we focus on test procedures which can rigorously incorporate this nonstandard and/or complex data into interpretable guarantees.

### A. Robot Policy Evaluation

**Evaluation Metrics.** Recent studies emphasize the importance of detailed rubrics and task progress scores [5], [1], [10] over traditional binary success metrics. In parallel, finetuning procedures [2] and policy ranking problems [21], [10] have introduced indirect signals of policy performance, including reinforcement learning (RL) rewards and human preferences. However, none of the approaches give rigorous methods for decision-making. Whereas the state-of-the-art procedure for policy comparison under binary success metrics [7] is rigorous and sample-efficient, current methods for more informative signals are not. This paper introduces a novel test procedure that scales rigorous evaluation to general metrics.

### B. The Parametric Policy Comparison Problem

The *parametric* setting for policy comparison arises when the evaluator designs the performance measure to have definite distributional structure (e.g., binary success or discrete partial credit). This structure specifies a tradeoff between the generality, sample efficiency, and correctness of the comparison.

Classic tests [22], [23], [24] were designed to optimize power for Bernoulli (binary) outcomes. Some subsequent results use tools from sequential analysis [25] to improve the sample efficiency [26], [27], [28]. In this regime, the STEP procedure [7] is state-of-the-art. Other developments extend the generality of these tests to broader classes of parametric distributions [29], [8]. However, simultaneously extending both parametric generality and sample efficiency is difficult, requiring an exact solution of the Wald-Bellman partial differential equation [30], [28], [31]. For more complex parametric families, finding this solution quickly becomes intractable [32], [33].

This motivates work that extends generality by making approximations in the large-data regime. A canonical example is Welch's t-Test [34], which is often used for batch comparison [1], [35], [36]. Further *asymptotic* results [12] can extend generality [37], [38], [39] *and* improve sample efficiency [40], [41], at the cost of losing rigorous statistical assurances; validity is not guaranteed in the low-data regime.

### C. (Nonparametric) Safe, Anytime-Valid Inference Methods

Nonparametric methods seek to exchange some exploitable structure for generality in application. Within the batch regime, these are often termed 'distribution free,' as

in the case of conformal prediction and related techniques [42], [43], [44]. However, they have two downsides: limited capacity to represent the underlying data distribution, and limited generalization to sequential settings. Kernel density estimation (KDE) addresses the first constraint by directly constructing a representation of the data-generating process [45]; see [46] for an overview. Critically, KDE is more efficient in low-dimensional settings [47] because it can quickly adapt to structure in the data. However, alone, it is not generally valid in finite samples [48].

To address the second constraint, work in the line of safe, anytime-valid inference (SAVI) considers *sequentialization* of estimation and decision problems, which act in an *online fashion* [14]. A core benefit is safety in the face of p-hacking [49], or 'data dredging' [50], [14]. Additionally, though not parametric, the framework does allow for methodological fine-tuning to the particular problem setting (as in [51], [52], [8]). Of these approaches, the WSR method [15] is closest to our own, considering the problem of estimating the mean of a general class of random variables. Importantly, N-SCORE utilizes ideas from KDE to tailor a SAVI framework *specifically to the comparison problem*, achieving state-of-the-art generality and sample-efficiency.

### D. Baseline Methods

We introduce three relevant baselines, proceeding in order of increasing generality. The recently proposed STEP procedure [7] demonstrated SOTA performance in the setting of binary success metrics. However, STEP is tailored to this narrow, but important setting and does not generalize to more informative metrics. Another recent work proposed a safe, anytime-valid inference approach to parametric comparison problems, motivated by settings with discrete partial credit [8]. This method, termed $\theta$-SAVI to denote its parametric nature, is more general than STEP, as it retains validity for any parametric comparison setting (defined in Section I-B). As with STEP, $\theta$-SAVI cannot extend to nonparametric performance metrics such as continuous-valued progress scores. The most general evaluation procedure is the test dual to the WSR estimation method [15], which is also most similar among the baselines to our approach. Thus, the key comparison between WSR and N-SCORE will center on sample efficiency.

We discuss instances of performance differences between N-SCORE and related baselines.

### A. LBM 1.0 Success and Partial Credit

Now, we revisit the results in Table A.I to consider the implications for each evaluation procedure. Here, we observe that STEP is again optimal for binary success measures, consistent with the previous section. For this data, $\theta$-SAVI, N-SCORE , and WSR are equally effective in both binary and discrete partial credit evaluation on this dataset.

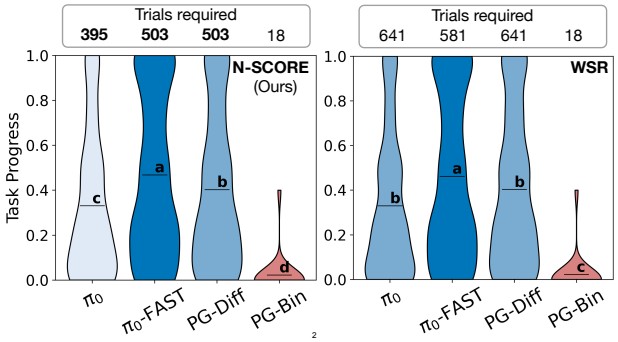

Fig. A.1: **Policy performance comparisons on crowd-sourced real-world evaluations on the DROID [53] setup from RoboArena [10]**. The violin plots represent empirical distributions of observed results. Policies with different letters are statistically distinguishable by the method. Policies are compared at a global error bound of $\alpha = 0.05$ with a Bonferroni correction.

### B. Results on RoboArena [10]

We continue the analysis of real-world evaluation data by illustrating the efficacy of N-SCORE in addition to its applicability to multi-policy comparisons using continuous progress evaluation scores, in simultaneously comparing four policies from the open-source RoboArena [10] benchmark. Because the rewards are continuous, only N-SCORE and WSR can be applied. Figure A.1 illustrates the time-to-decision (TTD) in terms of the number of trials required for each policy by N-SCORE and WSR. N-SCORE is able to distinguish the performance of all policies. In contrast, while WSR is able to correctly distinguish the best policy as $\pi_0$-FAST, it is unable to separate $\pi_0$ and PG-Diff even after exhausting all available 641 trials. Notably, N-SCORE requires over 200 fewer trials of $\pi_0$, and results in a total savings of at least **450 trials** (1419 vs. 1881). The efficacy of our method can be attributed to efficient optimization of $\xi_n$ (see Remark 1) with the available data, and due to WSR not being optimized for policy comparison.

As demonstrated by these extensive empirical validations, the key impact of our novel approach lies in effectively matching the sample efficiency of $\theta$-SAVI in parametric contexts (e.g., Table I and Table A.I), while maintaining the generality of, and improving sample efficiency over, the WSR procedure in nonparametric contexts.

## APPENDIX III
## ANALYTICAL SAVI RESULTS

In this section, we present the proofs for key theoretical results in the main text.

### A. Proof of Lemma 1

We state an important lemma that is integral to Equation (5):

*Lemma 1 (Null Stability (NSM) Property):* Consider the stochastic process $\{X_n\}$ defined in Equation (3), setting (w.l.o.g.) $X_0 = 1$. Then the expectation of $\{X_n\}$ is contracting in time with respect to the current value, for all $h \in S^-$

| Method → | Progress Metrics | | Binary Success/Failure Metrics | | | |
|---|---|---|---|---|---|---|
| Task ↓ | N-SCORE$_\infty$ | WSR | STEP | N-SCORE$_2$ | $\theta$-SAVI | WSR |
| DumpVegetables | 38 | 36 | 154 | – | – | – |
| PutContainer | 33 | 36 | 169 | – | – | – |
| PutFruit | 10 | 10 | 40 | 46 | 45 | 52 |
| SeparateFruit | 18 | 20 | 77 | 98 | 98 | 103 |
| TurnUpsideDown | – | – | – | – | – | – |
| Total (Simulation) | 598 | 604 | 1280 | 1488 | 1486 | 1510 |
| BikeRotor | – | – | – | – | – | – |
| CutApple | 29 | 29 | – | – | – | – |
| CleanLitter | 36 | 23 | – | – | – | – |
| ClearCounter | 16 | 15 | 23 | 25 | 30 | 46 |
| SetUpBreakfast | 12 | 13 | 15 | 19 | 19 | 17 |
| Total (Hardware) | 286 | 260 | 376 | 388 | 398 | 426 |

TABLE A.I: **Time-to-decision for all simulation (top) and hardware (bottom) policy comparisons for evaluation context in [1]**. If a decision is not reached, the entry is left blank; for the purpose of computing evaluation savings, any blank entry is counted at $N$ trials. All simulation tasks utilize $N = 200$; all hardware tasks utilize $N = 50$. The total number of trials is the column sum multiplied by two (because there are two policies being evaluated, and each must be run). Thus, a column in the top half could require up to 2000 simulated trajectories; one in the bottom half could require up to 500 hardware evaluations. Several important observations are: (1) sequential evaluation saves significant evaluation effort over batch methods (right); (2) more informative partial credit metrics can provide *even greater* savings (left). In fact, for these evaluations, the savings are up to 50% on hardware and 70% in simulation.

for any $\xi_n \in [0,1]$. That is:

$$\sup_{h \in S^-} \sup_{\xi_n \in [0,1]} \mathbb{E}_{r_0, r_1 \sim h}\left[ \frac{X_{n+1}}{X_n} \middle| \mathcal{F}_n \right] \le 1. \quad (6)$$

The proof proceeds directly. Consider the stochastic process increment in Equation (6), interpreted as the 'approximate marginal evidence increment' represented in Equation (3). We need to show that:

$$\sup_{h \in S^-} \sup_{\xi \in [0,1]} \mathbb{E}_{r_0, r_1 \sim h}(1 + \xi(r_1 - r_0)) \le 1. \quad (7)$$

The property arises directly out of the boundedness assumption of general progress metrics and the linear separability of $r_0$ and $r_1$ in the increment computation. From boundedness on $r_0$ and $r_1$, we know that the increment is bounded w.p. 1 in $[1 - \xi, 1 + \xi] \subseteq [0, 2]$. Therefore, the increment has well-defined moments.

The exact value can be computed as:

$$\mathbb{E}[1 + \xi(r_1 - r_0)] = \int 1 + \xi(r_1 - r_0) d\mu_{0,1}$$
$$= \int_0^1 \int_0^1 1 + \xi r_1 - \xi r_0 d\mu_0 d\mu_1$$
$$= \int_0^1 1 + \xi r_1 - \xi \mathbb{E}_{D_R^{[0]}}[R] d\mu_1$$
$$= 1 + \xi \mathbb{E}_{D_R^{[1]}}[R] - \xi \mathbb{E}_{D_R^{[0]}}[R]$$
$$= 1 + \xi(\mathbb{E}_{D_R^{[1]}}[R] - \mathbb{E}_{D_R^{[0]}}[R])$$
$$\le 1 \Longleftarrow (\mathbb{E}_{D_R^{[1]}}[R] - \mathbb{E}_{D_R^{[0]}}[R]) \le 0.$$

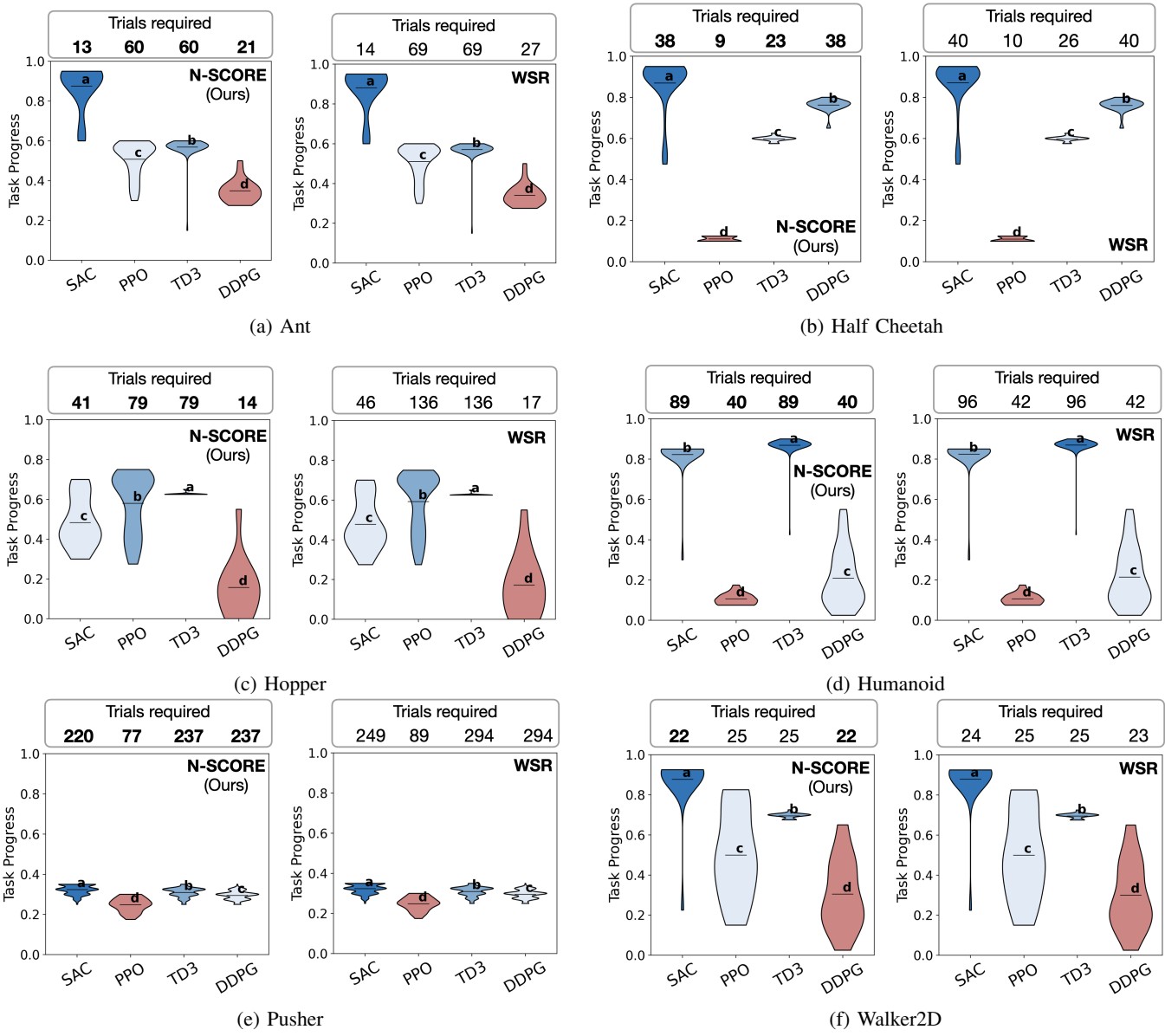

Fig. A.2: **Violin plots and the number of samples required for N-SCORE and WSR on multi-policy comparison of RL policies on Mujoco benchmarks.** Policies with different letters are statistically distinguishable by the method. Policies are compared at a global error bound of $\alpha = 0.05$ with a Bonferroni correction. In all cases, N-SCORE results in the same comparison conclusions as WSR with fewer samples, demonstrating its broadly improved efficiency. These results also serve as an alternate visualization of the time-to-decision results.

The last term is precisely the definition of $S^-$; therefore, we have shown the nonnegative (super)martingale property holds precisely for any instance in which the null hypothesis is true.[2] This argument directly extends to any parametric progress metric setting, due to the nature of the increment construction. Thus, it holds for binary, discrete, and continuous valued bounded metrics.

*B. Proof of Theorem 1*

We now utilize the results of Section III-A, which demonstrated the equivalent of the nonnegative supermartingale (NSM) property of the evidence aggregation *when the true state of the world lies within the set* $S^-$, to prove Theorem 1.

*1) Verifying Ancillary Conditions:* Note that by construction, $\xi_n \in [0, 1]$ and $(r_{1,n} - r_{0,n}) \in [-1, 1]$ implies that

$$\Delta_n = \left[1 + \xi_n(r_{1,n} - r_{0,n})\right] \in [0, 2] \text{ w.p. 1.}$$

---

[2]If we restrict $\xi \in (0, 1)$, then the relation holds bidirectionally, i.e., with $\Longleftrightarrow$.

| Comparison → | **PPO vs. DDPG** | | **SAC vs. TD3** | |
| Task ↓    Method → | WSR | N-SCORE$_\infty$ | WSR | N-SCORE$_\infty$ |
| --- | --- | --- | --- | --- |
| Ant-v4 | 27 | 21 | 14 | 13 |
| HalfCheetah-v4 | 8 | 8 | 18 | 15 |
| Hopper-v4 | 13 | 12 | 30 | 20 |
| InvertedPendulum-v4 | 677 | **267** | – | – |
| Humanoid-v4 | 42 | 40 | 96 | 89 |
| Walker2d-v4 | 23 | 22 | 24 | 22 |
| Pusher-v4 | 89 | 77 | 249 | 220 |
| Total (14000 nominal) | 1758 | **894** | 2862 | **2758** |

TABLE A.II: **Time-to-decision for selected reinforcement learning policy comparisons on Mujoco benchmarks.** If a decision is not reached, the entry is left blank; for the purpose of computing evaluation savings, any blank entry is counted at $N$ trials. All simulated tasks utilize $N = 1000$. We observe similar behavior between N-SCORE and WSR on easier instances (with lower times-to-decision); however, in harder instances significant improvements can be observed. In aggregate, the hard instances dominate sample complexity, resulting in substantial savings in evaluation burden.

| Task | DDPG | TD3 | PPO | SAC |
| --- | --- | --- | --- | --- |
| Ant-v4 | 418.8 | 2477.5 | 1849.0 | 4901.8 |
| HalfCheetah-v4 | 9932.2 | 7782.5 | 1332.2 | 11759.0 |
| Hopper-v4 | 1106.4 | 3234.6 | 2797.4 | 2488.7 |
| InvertedPendulum-v4 | 927.2 | 998.6 | 848.2 | 1000.0 |
| Pusher-v4 | -38.5 | -35.7 | -47.6 | -32.4 |
| Walker2d-v4 | 1732.5 | 3513.4 | 1679.8 | 4544.1 |
| Humanoid-v4 | 1386.0 | 5288.2 | 748.4 | 5042.7 |

TABLE A.III: **Empirical mean episodic return on evaluation instances for Mujoco benchmark tasks.**

Therefore,

$$X_n := X_0 \prod_{i=1}^{n} \Delta_i \ge 0 \ \forall n \ge 1. \tag{8}$$

This verifies nonnegativity.

*2) Ville's Inequality:* Ville's Inequality is the critical mechanism whereby the expectation of a nonnegative supermartingale process (see Definition 1) can be linked to right-tailed quantiles of its realized behavior.

*Definition 1 (Nonnegative Supermartingale (NSM)):*
Consider a discrete-time stochastic process $\{X_n\}_{n\ge 0}$ equipped with the natural filtration $\mathcal{F}_s = \{X_i\}_{i=0}^{s-1}$,[3] and w.l.o.g. let $X_0 = 1$. The process $\{X_n\}_{n\ge 0}$ is a **nonnegative supermartingale (NSM)** if it is everywhere nonnegative and contracting in expectation with respect to the filtration:

$$\inf_n\{X_n\}_{n\ge 0} \ge 0 \text{ w.p. } 1$$
$$\forall n, \ \mathbb{E}[X_n|\mathcal{F}_n] \le X_{n-1} \tag{9}$$

Intuitively, an NSM is a 'stable process' in that it is lower bounded by 0 and contracting in expectation. This stability is the intuitive mechanism from which Ville's Inequality arises.

*Theorem 2 (Ville's Inequality [54]):* Let $\{X_n\}_{n\ge 0}$ be a nonnegative supermartingale. Then for any $\alpha \in (0,1)$,

$$\mathbb{P}\left[\exists n \in \mathbb{N} : X_n \ge \frac{\mathbb{E}[X_0]}{\alpha}\right] \le \alpha. \tag{10}$$

The critical interpretation of this result is that, for all $n \ge 0$, the $1 - \alpha$ quantile of $X_n$ is upper bounded by $\mathbb{E}[X_0]/\alpha$ for any $\alpha \in (0,1)$. Importantly, this means that the result holds even for optional (i.e., selective) determination of a time of decision.[4]

*3) Time-Varying $\xi_n$:* We now confirm that the use of time-varying $\xi_n = g(\mathcal{F}_{n-1})$, measurable with respect to the filtration, do not violate the Type-1 Error bounds in Section III-A. This follows from the definition of the natural filtration and the independence of marginal evaluation outcomes. Specifically, we must modify the proof in Section III-A to account for the conditional dependencies of $\xi_n = g(\mathcal{F}_{n-1})$ on the previous data. However, the change is minimal due to The independence of $\xi_n$ with the *new data* $(r_{0,n}, r_{1,n})$. We include the modification for completeness:

$$\mathbb{E}[1 + \xi_n(r_{1,n} - r_{0,n})] = \mathbb{E}[1] + \mathbb{E}[\xi_n(r_{1,n} - r_{0,n})]$$
$$= \mathbb{E}[1] + \mathbb{E}[\xi_n]\mathbb{E}[(r_{1,n} - r_{0,n})]$$
$$= 1 + \mathbb{E}[\xi_n](\mathbb{E}_{D_R^{[1]}}[R] - \mathbb{E}_{D_R^{[0]}}[R])$$
$$\le \arg\max\{1, 1 + (\mathbb{E}_{D_R^{[1]}}[R] - \mathbb{E}_{D_R^{[0]}}[R])\}$$
$$\le 1 \Longleftarrow (\mathbb{E}_{D_R^{[1]}}[R] - \mathbb{E}_{D_R^{[0]}}[R]) \le 0.$$

The third line follows from the independence of $\xi_n$ and $(r_{0,n}, r_{1,n})$, and the maximum in the penultimate line arises from taking the extremal values (0 and 1) of $\mathbb{E}[\xi_n]$. We again observe that membership in the the null hypothesis is precisely sufficient to ensure the NSM property.

*4) Completing the Proof:* We consider the stochastic process defined in Equation (8) taking $X_0 = 1$. From Lemma 1 and the ancillary verification, we have that $\{X_n\}$ is a nonnegative supermartingale on $S^-$. Using Ville's Inequality, we conclude that, for any possible true state of the world represented by some $h \in S^-$, the probability that $\max_n\{X_n\}$ exceeds $1/\alpha^*$ is less than or equal to $\alpha^*$. Therefore, using the stopping rule defined in Equation (4), the probability of falsely rejecting any true null $h \in S^-$ is uniformly bounded by $\alpha^*$. This is equivalent to the claim of Equation (5).

### C. Proof of Remark 1

We describe in more detail the explicit nonparametric representation of the optimization problem for selecting $\xi_n = g(\mathcal{F}_{n-1})$. As described in Section IV, we draw inspiration from kernel density estimation to explicitly model the distribution of outcomes of new evaluation draws. We use a simple version KDE with a preset, uniform binning scheme. That is, we represent the distribution of evaluation scores for each policy $\pi_i$ with $k$ bins partitioning the interval $[0, 1]$. For the case of exact partial credit evaluation with $K$

---

[3]The natural filtration in this context is simply the available information on which a causal algorithm may act. Consistent with this semantic meaning, we note for completeness that $\mathcal{F}_0 = \{\emptyset\}$.

[4]In the language of stochastic processes and sequential analysis, the time of decision is often referred to as the "stopping time" of the process. Adaptively selecting to stop and decide or to continue collecting data is then referred to as "optional stopping."

outcomes, these bins can be chosen to precisely model the true underlying distribution when $k = K$. For nonparametric instances or cases with continuous densities, the choice of $k$ trades off greater accuracy in the representation ($k \uparrow$) against computational burden (which decreases as $k \downarrow$).

Importantly: when constructing the martingale increments in Equation (3) the exact (possibly continuously-valued) evaluation scores must be used to certify Type-1 Error control. However, no restriction is made with regard to how said data is used to sequentially construct $\xi_n$. Very informally, the algorithm to select the multiplier may 'deceive itself' however it likes without violating rigorous validity – it will simply risk being less efficient. This is precisely the key insight – N-SCORE will 'pretend' that the data is parametric partial credit (via discretization of the observed performance scores) when choosing $\xi_n$, allowing for efficient optimization. Nonetheless, this discretization of the observed data for selecting $\xi_n$ does not invalidate Lemma 1; it can only affect the efficiency of the process as measured by time-to-decision. This is in *stark contrast with* $\theta$-SAVI, which requires that the *true underlying distribution* be of a parametric (i.e., partial credit) form.

As a concrete example of the binning procedure, utilizing eleven bins, we may sort the data by its first two significant digits. This is equivalent to

$$\text{bin\_index}(r_{i,n}) = \lfloor 10 \cdot r_{i,n} \rfloor \in \{0, 1, \dots, 10\}. \quad (11)$$

This can then be seen as a lossy compression of the observed data, where we only represent its approximate value:

$$\tilde{r}_{i,n} \leftarrow \frac{\lfloor 10 \cdot r_{i,n} \rfloor}{10} \in \{0, 0.1, 0.2, \dots, 1.0\}. \quad (12)$$

The importance of this compression lies in reducing all (highly complex) distributions over general progress metrics to the (parametric) family of categorical distributions over $k$ outcomes, where $k$ is precisely the number of bins. Now, we fix the binning procedure to be shared for both policies, and denote the vector of $k$ compressed outcomes to be $\mathbf{c} \in \mathbb{R}_+^k$. We will w.l.o.g. assume henceforth that $\mathbf{c}$ is ordered from least to greatest, and that the elements $\mathbf{c}_i$ are distinct.[5] The true underlying distributions over outcome scores are compressed to vectors on the $k$-simplex:

$$r_{i,n} \sim \mathcal{D}_R^{[i]} \implies \tilde{r}_{i,n} = \mathbf{c}_j; j \sim \text{Categorical}(\mathbf{p}_i). \quad (13)$$

There are various useful quantities which arise from this representation. The probability of each possible joint evaluation outcome (i.e., of $\pi_0$ and $\pi_1$) can be simultaneously represented as a $k \times k$ square matrix $P$:

$$P = \mathbf{p}_0 \mathbf{p}_1^T, \quad (14)$$

where $P_{ij}$ is understood to be 'the probability that, for a new evaluation draw, $\pi_0$ gets a (compressed) score $\tilde{r}_{0,n} = \mathbf{c}_i$

and $\pi_1$ gets a (compressed) score $\tilde{r}_{1,n} = \mathbf{c}_j$.' Furthermore, the set of approximate evidence integrator outcomes can be represented by a $k \times k$ square matrix $A$, where:

$$A_{ij}(\xi_n; \mathbf{c}) = 1 + \xi_n(\mathbf{c}_j - \mathbf{c}_i). \quad (15)$$

To link this to Lemma 1: in the special case of discrete partial credit structure, Lemma 1 amounts to demonstrating that the following statement is true *under the null*:

$$\sup_{\mathbf{p}_i} \langle A, \mathbf{p}_0 \mathbf{p}_1^T \rangle \leq 1$$
$$\text{for all} \quad \mathbf{p}_i \in \Delta_k \quad (16)$$
$$\mathbf{c}^T(\mathbf{p}_1 - \mathbf{p}_0) \leq 0.$$

Conversely, Lemma 1 is sufficient to demonstrate that the above statement must be true, as the latter follows from the generality of the former.[6] The key point here is that this is precisely a stability condition on the expectation of the evidence aggregator *when the true state of the world is an element of the null set* $S^-$. The optimization of $\xi_n$, by contrast, relates to optimally *de-stabilizing* the evidence aggregator when the the true state of the world is an element of the alternative set, $S^+$. In that setting, we very much wish for the expectation to be *greater than one*, in contrast with Equation (16).

*1) Intuition for Optimizing $\xi$:* With the preceding development, our KDE-inspired approach attempts to optimize $\xi$ over the lossy representation induced by the discretization (i.e., the nonparametric distribution representation as, approximately, a discrete partial credit random variable).[7] Unlike in verifying the NSM property (Lemma 1), the linear algebraic representation of the partial credit problem provides insight into choosing $\xi_n$. Recall that the choice of $\xi_n$ does not affect Type-1 Error, and therefore does not affect any state of the world in which the null is true. Therefore, it will only be used to accelerate detection when the state of the world is such that the alternative is true. The core idea is to observe two phenomena arising out of realizations of $A_{ij}$: the 'signal effect' and the 'hysteresis effect.'

*2) Signal Effect:* The 'signal effect' amounts to direct evidence for the alternative. This arises when $r_{1,n} > r_{0,n}$; when this is the case, the multiplier $\Delta_n$ grows with $\xi_n$. Thus, greater likelihood of seeing positive differences in the metrics ('positive signals') promotes *increasing* the value of $\xi$. This is linear in the *asymmetric component* of $P$, in the following sense. Define

$$\Delta P_{ij} := P_{ij} - P_{ji}, \text{ for } j > i, \quad (17)$$

and 0 otherwise. By definition, $\Delta P$ is an upper triangular matrix. In general, if the matrix has more *positive* elements, this is evidence that the alternative is more likely to be true. That is, $\Delta P_{ij} > 0$ for some $i < j$ means that the probability of observing $r_{1,n} - r_{0,n} = \mathbf{c}_j - \mathbf{c}_i > 0$ is *larger than* the

---

[5]Distinctness is not restrictive; if any set of (semantic) outcomes has the same evaluation score, then they can be 'lumped together' into a single composite outcome. The binning procedure itself is assumed to be a deterministic function of the evaluation outcome; therefore, it will always ensure distinction between outcomes it observes. The particular semantic meaning of a score, however, may not be directly observable.

[6]This can be shown independently using properties of the matrix $A$ and some linear algebraic identities, but is outside the scope of the core intuition.

[7]Doing this ever-more efficiently is precisely a subject of future work, as much of the specific domain knowledge of KDE is not present in our simplified implementation.

converse of $r_{1,n} - r_{0,n} = \mathbf{c}_i - \mathbf{c}_j < 0$. Considering all pairs $(i, j)$, we observe that $\Delta P$ precisely encodes *asymmetry* in $P$ and its contribution to differences in the mean performance of $\pi_0$ vs $\pi_1$. Considering the expected martingale growth rate, we can observe now that this contributes to the growth linearly in $\xi$:

$$\mathbb{E}\left[\frac{M_{n+1}}{M_n}\bigg|(\tilde{r}_{0,n}, \tilde{r}_{1,n}) = (i, j), j > i\right]$$
$$= \left((1 + \xi(\mathbf{c}_j - \mathbf{c}_i))\Delta P_{ij}\right) \qquad (18)$$
$$= 1 + \xi(\mathbf{c}_j - \mathbf{c}_i)\Delta P_{ij}.$$

Therefore, the aggregate positive evidence that $\pi_1$ is better than $\pi_0$ is the sum of these pairwise effects:

$$\sum_{i=1}^{k}\sum_{j=i}^{k}(1 + \xi(\mathbf{c}_j - \mathbf{c}_i))\Delta P_{ij}.$$

As noted previously, the key idea of N-SCORE is to estimate the quantity $\Delta P$ from the data currently observed (i.e., the filtration $\mathcal{F}_n$) to choose an effective $\xi_n$.

*3) Hysteresis Effect:* The signal effect generally pushes $\xi_n$ to be larger; by contrast, there is an opposing mechanism which induces it to shrink. This relates to the *symmetric* component of the $P$ matrix, and is termed the 'hysteresis effect.' This effect is so named because of how it manifests: symmetric aspects of $P$ correspond to 'self-negating' outcomes (e.g., in which $\tilde{r}_{1,n} - \tilde{r}_{0,n} = -(\tilde{r}_{1,n-1} - \tilde{r}_{0,n-1})$). Direct inspection should convince the reader that difference in empirical performance between the policies has not changed from step $n-2$ to step $n$ (each has observed the same total return since step $n-2$). However, in the course of cycling through zero net change in mean performance difference, the value of $X_n$ has *decreased* from $X_{n-2}$. Achieving both of the converse outcomes (i.e., $(i, j)$ and $(j, i)$) is reflected in the *symmetric component* of $P$: the fraction of realizations of $A_{ij}$ which will be 'counteracted' by realizations of $A_{ji}$. As just stated, these pairs of outcomes do not change the empirical gap between the policies, but they *negatively impact* the $X_n$. The idea of losing value (in $X_n$) via a closed loop in net performance difference motivates the term 'hysteresis.' Mathematically, we first define

$$\underline{P}_{ij} = \min\{P_{ij}, P_{ji}\}. \qquad (19)$$

This symmetric matrix quantifies the degree to which hysteresis plays a part. The effect on the stochastic process value can be observed via approximate Taylor Expansion (assuming for now that $\xi_n$ is slowly varying):

$$X_{n+2} = X_n(1 + A_{ij})(1 + A_{ji})\underline{P}_{ij}$$
$$\implies X_{n+2}/X_n = (1 + \xi_n(\mathbf{c}_j - \mathbf{c}_i))(1 - \xi_{n+1}(\mathbf{c}_j - \mathbf{c}_i))\underline{P}_{ij}$$
$$\approx (1 + \bar{\xi}(\mathbf{c}_j - \mathbf{c}_i))(1 - \bar{\xi}(\mathbf{c}_j - \mathbf{c}_i))\underline{P}_{ij}$$
$$= 1 - \bar{\xi}^2(\mathbf{c}_j - \mathbf{c}_i)^2\underline{P}_{ij}. \qquad (20)$$

This term acts to regularize the choice of $\xi_n$, because it suggests that, even with no net gain of information, the stochastic process will tend to decay, and that this decay is larger when $\xi_n$ is larger. Thus, hysteresis motivates a smaller $\xi_n$, opposing the signal effect. However, importantly, unlike the signal effect, the hysteresis effect is quadratic in $\xi$. At an informal level, this is suggestive of maximizing a concave quadratic function over a convex domain, which is a convex optimization problem. Continuing the informal discussion, this suggests that the signal effect (which is linear) will always locally dominate and $\xi_n$ will never be forced to zero when the means differ favorably.

*Remark 2 (Linearity of $\underline{P}_{ij}$):* The optimization of $\xi_n$ is implicitly single-step, which should not be suboptimal given the temporal independence of evaluation outcomes. The weighting of the hysteresis terms arises from understanding each component in the single-step context. That is, evaluation outcomes are partitioned as "an observation $(i, j)$ which will be balanced out by an associated $(j, i)$" (hysteresis) versus "an observation $(i, j)$ which will *not* be balanced out by an associated $(j, i)$" (signal). Of course, the outcomes which will cancel in the future are twice $\underline{P}_{ij}$ (because one can get either the contributing $(i, j)$ outcome *or* the $(j, i)$ outcome), but the fact that *both* outcomes are required to achieve hysteresis means that each individual observation (that is, $(i, j)$ xor $(j, i)$) should be weighted *by one half*. Thus, the appropriate weighting in the right-hand term in Equation (20) is precisely $\underline{P}_{ij}$, as opposed to either $\underline{P}_{ij}^2$ (which is not single-step) or $2\underline{P}_{ij}$, which does not take into account the fact that *both outcomes* are needed for hysteresis to occur.

*4) The Optimization Problem:* With the preceding development, we attempt to maximize the log-value of the stochastic process via single-step optimization, given the currently available information.

Breaking this down: the log-value of $X_n$ is precisely the sum of the log $\Delta_i$, per the definition of the evidence integrator in Section IV. The available information manifests as

$$\hat{P} = \hat{\mathbf{p}}_0\hat{\mathbf{p}}_1^T, \qquad (21)$$

which represents the empirical distribution of the observed results. Each $\hat{\mathbf{p}}_i$ can be computed by simply calculating the empirical frequency of occurrence of each bin. Thus, the approximate expected multiplier value (given the current available information) is $\langle A, \hat{P}\rangle$. Lemma 1 guarantees that this expectation is bounded by 1 when the null is true; we are interested, however, precisely in the case where the alternative is true. In that setting, the bound does not apply.

Due to the linearity of the inner product, the optimization essentially acts elementwise on $\hat{P}$-weighted elements of $\log A_{ij}$:

$$\xi^* = \arg\max_{\xi \in [0,1)} \left(\sum_{i=1}^{K-1}\sum_{j=i+1}^{K}|\Delta\hat{P}_{ij}|\log\left(1 + \xi \operatorname{sgn}(\Delta\hat{P}_{ij})\Delta c_{ij}\right)\right.$$
$$\left. \ldots + \hat{\underline{P}}_{ij}\log\left(1 - \xi^2\Delta c_{ij}^2\right)\right).$$

Applying first-order optimality conditions, we obtain an efficient representation for which a root-finding procedure

on $[0, 1]$ quickly converges:

$$0 = \nabla_\xi(\ldots)|_{\xi^*}$$
$$= \sum_{i=1}^{K-1} \sum_{j=i+1}^{K} \frac{\Delta \hat{P}_{ij} \Delta c_{ij}}{1 + \xi^* \operatorname{sgn}(\Delta \hat{P}_{ij}) \Delta c_{ij}} - 2\xi^* \frac{\hat{P}_{ij} \Delta c_{ij}^2}{1 - (\xi^*)^2 \Delta c_{ij}^2}.$$

This can be understood as choosing the optimal $\xi_n$ to maximize the expected growth rate of $X_n$ under the currently available estimate of the distributions $\hat{\mathbf{p}}_i$. This choice explicitly balances the signal and hysteresis effects in order to achieve this maximization. Further, though it is beyond the scope of the present work, it is believed that the nominal objective is in fact concave (via analysis of the second-order shape of the objective). Thus, the current approach is likely sufficient despite only the first-order verification; additionally, faster approaches are likely feasible. Regardless, this proves the necessary result, and describes the practical optimization scheme used for N-SCORE in all experiments.