# OpenReview forum: "Sample-Efficient and Statistically Rigorous Policy Comparison Using General Robot Performance Measures"
_IEEE.org/ICRA/2026/Workshop/Manipulation_Robustness — ICRA 2026_

### Official Review · Reviewer_2B5g · 2026-05-01
**Useful sequential test for nonparametric policy comparison; recommend accept**

**Rating:** 7
**Confidence:** 4

**Review:**

# Strengths

The paper targets a real bottleneck in robot policy evaluation: hardware rollouts are scarce, and the existing rigorous tools (STEP for binary success, θ-SAVI for parametric partial credit) don't extend to the continuous, behavior-level signals that manipulation-robustness research increasingly relies on (progress scores, jitter, etc.). N-SCORE fills exactly that gap.

The technical contribution is well-executed. The SAVI / Ville's-inequality scaffolding is standard, but the KDE-style discretization for choosing ξ_n online is a genuinely good idea, and the resulting algorithm (Alg. 1) is short and easy to implement. Type-1 error control carries through cleanly (Theorem 1, Lemma 1), and the proofs in App. III are easy to follow.

The empirical results are convincing. On simulated nonparametric data N-SCORE is meaningfully more efficient than WSR (TTD 206.8 vs 247.3, p<0.05); on six Mujoco tasks it consistently saves trials across four RL algorithms; and on the DROID/RoboArena multi-policy comparison it saves on the order of 450 trials over WSR while still separating all four policies, which WSR cannot.

The workshop fit is strong. Sec. VI explicitly frames N-SCORE as evaluation infrastructure for future robustness measures rather than as a robustness method itself, which is the right framing for this venue.

# Weaknesses

The main paper is unusually compressed and leans heavily on the appendix — the hardware experiments, the Mujoco per-task breakdown, and all the proofs are out there. Surfacing at least one hardware result in the body would make the contribution easier to evaluate from the main text alone.

The optimization of ξ_n relies on a KDE-style discretization whose concavity is asserted but not proved ("believed... via second-order shape"). A full proof isn't expected at workshop length, but a brief empirical comparison against a simpler choice of ξ_n (e.g., a constant or fixed schedule) would help quantify how much of the efficiency gain over WSR actually comes from the online optimization vs. from the SAVI framing more broadly.

Practical guidance is also thin: how should the bin count k be chosen, and what is the recommended practice when the performance signal isn't already in [0,1]? These are the first decisions an evaluator using N-SCORE has to make, and the paper currently leaves them implicit.

# Overall comments

This is a focused, well-motivated contribution that closes a real gap in the policy-evaluation toolkit and aligns directly with the workshop's interest in measuring manipulation robustness. The theory is sound where it needs to be, the empirical case is consistent across simulated and real-world data, and the method is practical enough that practitioners could adopt it directly. The remaining weaknesses are mostly presentational and would be straightforward to address in revision. I recommend acceptance.

---

### Decision · Program_Chairs · 2026-05-21

Accept